# Modeling of the Aqueous Solubility of N-butyl-N-methyl-1-phenylpyrrolo[1,2-a] pyrazine-3-carboxamide: From Micronization to Creation of Amorphous–Crystalline Composites with a Polymer

**DOI:** 10.3390/polym15204136

**Published:** 2023-10-18

**Authors:** Vladimir B. Markeev, Sergey V. Tishkov, Anton M. Vorobei, Olga O. Parenago, Evgenia V. Blynskaya, Konstantin V. Alekseev, Anna I. Marakhova, Alexandre A. Vetcher

**Affiliations:** 1V.V. Zakusov Research Institute of Pharmacology, 8 Baltiyskaya St., 125315 Moscow, Russia; sergey-tishkov@ya.ru (S.V.T.); evaureus@gmail.com (E.V.B.); convieck@yandex.ru (K.V.A.); 2Kurnakov Institute of General and Inorganic Chemistry, Russian Academy of Sciences, 31Leninsky Pr., 119071 Moscow, Russia; vorobei@supercritical.ru (A.M.V.); oparenago@scf-tp.ru (O.O.P.); 3Institute of Biochemical Technology and Nanotechnology, Peoples’ Friendship University of Russia n.a. P. Lumumba (RUDN), 6 Miklukho-Maklaya St., 117198 Moscow, Russia; agentcat85@mail.ru

**Keywords:** polymorphism, micronization, solubility, polyvinylpyrrolidone, anxiolytics, antidepressants

## Abstract

N-butyl-N-methyl-1-phenylpyrrole[1,2-a] pyrazine-3-carboxamide (GML-3) is a potential candidate for combination drug therapy due to its anxiolytic and antidepressant activity. The anxiolytic activity of GML-3 is comparable to diazepam. The antidepressant activity of GML-3 is comparable to amitriptyline. GML-3 is an 18 kDa mitochondrial translocator protein (TSPO) ligand and is devoid of most of the side effects of diazepam, which makes the research on the creation of drugs based on it promising. However, its low water solubility and tendency to agglomerate prevented its release. This research aimed to study the effect of dry grinding, the rapid expansion of a supercritical solution (RESS), and the eutectic mixture (composite) of GML-3 with polyvinylpyrrolidone (PVP) on the particle size, dissolution rate, and lattice retention of GML-3. The use of supercritical CO_2_ in the RESS method was promising in terms of particle size reduction, resulting in a reduction in the particle size of GML-3 to 20–40 nm with a 430-fold increase in dissolution rate. However, in addition to particle size reduction after RESS, GML-3 began to show signs of a polymorphism phenomenon, which was also studied in this article. It was found that coarse grinding reduced particle size by a factor of 2 but did not significantly affect solubility or crystal structure. Co-milling with the polymer made it possible to level the effect of the appearance of a residual electrostatic charge on the particles, as in the case of grinding, and the increased solubility in the resulting mechanical mixtures of GML-3 with the polymer may also indicate the dissolving properties of polymers (an increase in 400–800 times). The best result in terms of GML-3 solubility was demonstrated by the resulting GML-3:PVP composite at a ratio of 1:4, which made it possible to achieve a solubility of about 80% active pharmaceutical ingredient (API) within an hour with an increase in the dissolution rate by 1600 times. Thus, the creation of composites is the most effective method for improving the solubility of GML-3, superior to micronization.

## 1. Introduction

Mitochondrial translocator protein (TSPO) is a promising target for the creation of new drugs with a wide spectrum of neuropsychotropic activity, including anxiolytic, antidepressant, nootropic, and neuroprotective [1,2]. TSPO (Mw 18 kDa) transports cholesterol from the outer to the inner membrane of mitochondria, which determines the rate of neurosteroid biosynthesis in the cells of the central nervous system. The strategy of pharmacological regulation of TSPO is the scientific basis for the creation of anxiolytics that have a pronounced superiority over benzodiazepines, whose pharmacotherapy is complicated by a complex of side effects: sedation, muscle relaxation, memory impairment, addiction, and withdrawal syndrome. The GML-3 molecule belonging to the 1-arylpyrrole[1,2-a]pyrazine-3-carboxamides group is a ligand of a TSPO [1,2]. Among those of the original derivatives of 1-arylpyrrole[1,2-a]pyrazine-3-carboxamides synthesized in the FSBSI Zakusov Research Institute of Pharmacology, it is GML-3 that showed pronounced anxiolytic and antidepressant activity in various animal tests, which makes it a promising compound for creating a drug with combined action [3]. It needs to be underlined that GML-3 is a novel compound.

The anxiolytic activity of GML-3 is comparable to diazepam. The antidepressant activity of GML-3 is comparable to amitriptyline. However, most benzodiazepines (including diazepam) cause side effects such as sedation, drug dependence, and withdrawal syndrome, which limits their clinical use. Namely, GML-3 from all of 1-arylpyrrolo[1,2-a]pyrazine-3-carboxamides’ derivatives simultaneously exhibits two therapeutic effects that are necessary for the treatment of depression. As a rule, patients need to take tranquilizers in addition to antidepressants. Taking several powerful medications at once can harm the patient’s body. Thus, GML-3, which is devoid of most of the side effects of diazepam and amitriptyline, can become a promising drug to combat depression. However, the creation of the oral form of the drug based on the API GML-3 may be hampered by the poor solubility of GML-3 in water. GML-3 belongs to the Class IV BCS.

The problem of water solubility is typical for 40% of drugs on the market and for more than 70% of new molecules approved by the Food and Drug Administration (FDA) [4,5,6]. According to the biopharmaceutics classification system (BCS), this can negatively affect bioavailability, and as a result, researchers have now developed many approaches to improve the solubilization and dissolution rates of molecules in water. These include particle size reduction, the creation of solid dispersions (API and polymer composites), the formation of complexes with cyclodextrins, self-emulsifying drug delivery systems (SEDDS) or liquid crystals (LCs), etc.

Micronization of pharmaceutical compounds is one of the most important processes in the pharmaceutical industry when it comes to increasing the dissolution rate of API. The micronization method makes it possible to increase the rate of expansion by increasing the area of contact with the medium. However, it is worth noting that the crystallinity of the API can be a limiting factor that slows down dissolution since, after micronization, it is rarely possible to obtain an amorphous API. Moreover, after micronization, it is possible to accumulate a residual static charge, which will contribute to the aggregation and recrystallization of particles during storage and contact with the medium. An alternative approach is based on changing the crystallinity by creating solid dispersions. The method is based on the creation of a composite of API and polymer, where API solubility is achieved due to the changed crystallinity of API and the solubilizing properties of the polymer. However, these composites may be unstable during their shelf life.

The traditional size of pharmaceutical powders, generally ranging from a few hundred to 50 μm, is an additional factor that reduces the dissolution rate of API, which negatively affects bioavailability, and therefore micronization is often required [7]. Based on the Noyes–Whitney equation, the dissolution rate is directly related to the contact area between the particles and the dissolution medium [8]. GML-3 is practically insoluble in water (more than 1:10,000) and is an agglomerate of crystals up to 200 µm in size. Given these circumstances, micronization was chosen as a method for increasing the dissolution rate. The crystals obtained by this method are called “nanocrystals” and usually refer to particles smaller than 1000 nm [9,10]. The promise of this approach is confirmed by the presence on the market of several drugs based on “nanocrystalline” technology [11,12,13].

In general, methods for obtaining nanocrystals can be divided into two groups: top–down methods and bottom–up methods. The first group includes grinding (dry and wet) and homogenization under high pressure, based on the principle of applying mechanical energy to break large crystals into smaller fragments [14,15,16,17]. As a rule, the resulting particles have an irregular shape, and the method itself has limitations on the size of the particles obtained, beyond which further grinding is impossible.

The second group includes methods of crystallization from solutions. The main technologies include spray drying, nanoprecipitation, RESS, and supercritical anti-solvent (SAS) deposition [18,19,20,21,22,23,24,25,26]. RESS technology can produce particles smaller than 100 nm, provided that the API is soluble in the supercritical fluid and does not degrade at high pressure. The API is dissolved in a supercritical fluid, often CO_2_, then depressurized through a nozzle into an expansion chamber, rapidly forming crystallization and fine particles. In the case of the insolubility of API particles in carbon dioxide, the SAS technology is used. Following this technology, the API is dissolved in an organic solvent, which in turn is soluble in supercritical carbon dioxide. This solution is injected through a nozzle into supercritical carbon dioxide, and the API is reprecipitated in a finely dispersed state.

It should be noted that thermal and mechanical effects on API crystals, as well as dissolution followed by crystallization, can change the polymorphic composition of a substance [27,28,29,30,31]. Polymorphism is the ability of the molecules of a substance to combine into various crystal lattices. The FDA pays increased attention to the potential effect of changing the polymorphic composition on changing the physicochemical properties of APIs [32,33,34,35]. Since this entails a change in the bioavailability and toxicity of the drug, there are several rules described in the guidance of the International Conference on Harmonization Q6A by Decision Tree No. 4, describing the requirements for controlling the behavior of the API in the solid state [36]. According to the European Pharmacopoeia, 57% of APIs can form hydrates, 20% form solvates, and about 58% are polymorphs [37,38]. Thus, the study of the presence of the phenomenon of polymorphism during recrystallization and the development of new molecules is mandatory.

Also, API after micronization may show a tendency to agglomerate due to residual electrostatic stress. This phenomenon can be neutralized by adding to the micronization process (both top–down and bottom–up) water-soluble polymers applicable in pharmaceutical technology, which are capable of providing supersaturation of the dissolved API in water, which is an additional positive factor in solving the problem of solubility. In the case of top–down, after adding the polymer to the grinding process, the mechanical mixture is usually formed (less often, eutectic with a common melting point). In the case of bottom–up, the polymer and API are usually dissolved in carbon dioxide or ethanol. If the API and polymer are dissolved well in this solvent after removal of the solvent at a certain ratio of API–polymer, a composite of the eutectic mixture type is formed, which is characterized by a common melting/glass transition temperature of the components of the composite when heated with partial preservation of crystallinity on X-ray powder diffraction (PXRD). It should be noted that the achievement of complete API amorphization by increasing the polymer content in the composite is evidence that crystallization has been inhibited, and the resulting composite is an amorphous solid dispersion (ASD). This method is an alternative to micronization and allows for a complete solution to the problem of crystallinity. However, there is a risk of recrystallization of the API during the creation, storage, and process of dissolving ASD in water. Thus, the degree of crystallinity and stability of ASD during storage should be controlled.

The creation of API composites with polymers is one of the most modern methods of improving solubility. The composite is also able to change the dissolution kinetics, which can improve the therapeutic effect. The use of water-insoluble polymers allows the use of a diffusion mechanism to prolong the release of API. However, each API requires analysis of crystallinity and changes in the release profile. This is due to the inability to pre-evaluate the effect of the method of obtaining ASD and polymer on the crystalline structure and kinetics of API release.

In this work, a study was conducted to obtain and evaluate the increase in the dissolution rate of micronized GML-3 obtained by various grinding methods. In connection with the possibility of the formation of polymorphic forms of API, a complex analysis of the resulting particles was carried out using Fourier-transform infrared (FTIR) spectroscopy, Raman spectroscopy, differential scanning calorimetry (DSC), scanning electron microscopy (SEM), and PXRD. Also, attention is paid to the creation of composites with PVP and polymerhydroxypropyl methylcellulose (HPMC) by top–down and bottom–up methods in the process of micronization and their influence on the dissolution rate of GML-3.

## 2. Materials and Methods

### 2.1. Materials

API: GML-3 (FSBSI Zakusov Research Institute of Pharmacology (Russian Federation, Moscow, Russia)). Release series: 30082021. Date of manufacture: 30 August 2021. Figure 1 shows the structural formula of API GML-3, and Table 1 shows its physical and chemical parameters.

Excipients (Es): Polymer polyvinylpyrrolidone (Kollidon^®^ 25; Mw 24–27kDa; Molar volume (MV) = 24,800 cm^3^/mol; Melting Point = 165 °C) was obtained from BASF AG, Ludwigshafen, Germany; HPMC (Methocel E4M CR Premium USP 2910; Mw 82–90kDa; MV = 115,000 (cm^3^/mol); Melting Point = 121 °C) was a kind gift from Colorcon Ltd., Dartford, UK.

Solvents: distilled water was obtained on the PE-2205 apparatus (Ecroskhim Ltd., St. Petersburg, Russia); ethanol HP (99.5%, 0.005% maximum water) was obtained from Merck KGaA AG, Darmstadt, Germany; CO_2_pur was obtained from Dorogobuzh Ltd., Smolensk, Russia.

### 2.2. Methods

#### 2.2.1. Dry Grinding

In an agate mortar (mortar outer diameter: 50 mm, mortar inner diameter: 40 mm, mortar depth: 13 mm, pestle length: 50 mm, pestle diameter: 11 mm, mortar volume: 10 mL, Xiamen TOB New Energy Ltd., Xiamen, China), 200 mg of GML-3 were subjected to rough mechanical impact for 20 min.

#### 2.2.2. Grinding with Polymer

In an agate mortar (mortar outer diameter: 50 mm, mortar inner diameter: 40 mm, mortar depth: 13 mm, pestle length: 50 mm, pestle diameter: 11 mm, mortar volume: 10 mL, Xiamen TOB New Energy Ltd., Xiamen, China), 100 mg of GML-3 and 1000 mg of PVP or HPMC were subjected to rough mechanical action with mixing for 20 min.

#### 2.2.3. Creating a GML-3 Eutectic Mixture with Kollidon 25

GML-3 (1.00 g) was loaded into a closed container (50 mL) and dissolved in 99% ethanol (30.00 g). The dissolution was carried out using a magnetic stirrer PE-6100 (Ecroskhim Ltd., St. Petersburg, Russia). Kollidon^®^ 25 was gradually added to the resulting solution and mixed until a homogeneous, transparent solution was obtained. The resulting solution was dried at 55 °C for 12 h.

#### 2.2.4. Recrystallization by Rapid Expansion of Supercritical Solution (RESS)

The pressure in the 500 mL expansion vessel was atmospheric. Carbon dioxide was used as a fluid. The following state parameters were used: Pressure: 5–20 MPa, temperature: 45–55 °C. The scheme of the RESS installation is shown in Figure 2.

#### 2.2.5. Characterization of Samples

The samples obtained were analyzed by PXRD, FTIR spectroscopy, Raman spectroscopy, DSC, and laser diffraction. The microscopic structure of the samples was observed using scanning electron microscopy at an accelerating voltage of 20 kV on a ZEISSE VOMA 10 apparatus (Carl Zeiss AG, Jena, Germany). PXRD patterns were collected on AN X Bruker AXS D8 Advance diffractometer (Bruker Corp., Karlsruhe, Germany) using a Ni filter, CuKa radiation, a voltage of 40 kV, and a current of 30 mA. The scanning rate was 3 s/step over a 2θ interval of 2–50°. DSC was performed using a Netzsch STA 449 F1 differential scanning calorimeter coupled with a QMS 403 C thermogravimeter (Netzsch Instruments North America LLC, Burlington, MA, USA). Approximately 5 mg of the sample was weighed and sealed in an aluminum container with a hole in the lid. The samples were heated to 160 °C at a heating rate of 5 °C/min. Air was used as the purge gas. The FTIR spectrum was recorded on an ATP 8900 plus instrument (DLaTGS detector, diode laser, standard spectral range 350–7800 cm^−1^), and the Raman spectrum on an ATR 6500 instrument (CCD detector type, laser wavelength 785 nm) was provided by Optosky Optical Ltd., Xiamen, China. Laser diffraction was carried out by the dry dispersion method on a Bettersizer 2600 instrument (Dandong Bettersize Instruments Ltd., Dandong, China).

#### 2.2.6. Dissolution Test

The dissolution test was carried out according to the European Pharmacopoeia 10 edition (2.9.3. Dissolution test for solid dosage forms). The medium of dissolution was distilled water with a volume of 900 mL. Mixing was carried out using a paddle mixer at a rate of 50 rpm. Dissolution medium temperature: 37.0 ± 0.5 °C; sampling time: at 1, 3, 5, 10, 15, 30, 45 and 60 min; replenishing the medium after each sampling (10 mL). The optical density was determined using a spectrophotometer PE-5400UF (Ecroskhim Ltd., St. Petersburg, Russia) at a wavelength of λ = 252–256 nm.

Comparison solutions were used to select the wavelength and evaluate the results of spectrophotometry. The comparison solution: the exact suspensions of GML-3 were placed in a measuring flask with a capacity of 50 mL. Then, they were dissolved in a sufficient amount of ethanol. Next, the volume of the solution in the flask was brought to the mark with the same solvent (water, alcohol solution, etc.) and mixed. A total of 1 mL of the resulting solution was transferred to a measuring flask with a capacity of 100 mL, bringing the volume of the solution in the flask to the mark with a solution of the mixture mixed. The optical density of the tested solutions and the comparison solution was measured on a spectrophotometer using an ethanol mixture as a comparison solution: experiment environment (1:100). According to the spectrophotometer data, it is optimal to make measurements for water and acetate buffer at 252 nm, alcohol solutions at 256 nm, and 0.01 M HCL at 254 nm.

#### 2.2.7. Accelerated Aging Stability Tests

Samples were stored in closed containers at 55 ± 0.5 °C for 175 h, which approximately corresponds to 3 months of storage at room temperature.

## 3. Results and Discussion

### 3.1. Particle Size of GML-3

At the first stage, the shape and characteristic particle size, crystallinity, and melting point of GML-3 were studied. This made it possible to evaluate the need for micronization of the API, the degree of crystallinity, and thermal stability when heated for subsequent dissolution in CO_2_.

The particle size was measured using a laser diffractometer equipped with a dynamic image analysis system (Figure 3d). In dynamic image analysis, a large number of particles move past the camera system and are analyzed in real-time. Using this system, the degree of sphericity and the ratio of length to width (mainly large particles) were estimated. Only 20% of the particles had a degree of roundness close to 1. When analyzing the length (L) of the particles to the width (D), a sufficiently large L/D spread was observed (30% of the particles are conditionally square). Thus, the particles have an irregular shape, which is also confirmed by the incorrect morphology of individual particles recorded by SEM (Figure 3a–c).

The average size of GML-3 is 58.64 µm, and particles larger than 200 µm are completely absent. The Gaussian distribution of particles is asymmetric, with a disproportionately large content of particles less than 30 µm (Figure 3a–d). This is due to the fact that the Fraunhofer theory is poorly applicable to measuring particles smaller than 50 µm. Residual static stress when working with the API was not visually observed. According to PXRD, at normal conditions, GML-3 is a crystal with characteristic peaks at 5.7°, 7.5°, 12.1°, 17.9°, 21.1°, and 22.7° (Figure 3e). GML-3 is completely in the crystalline state without the inclusion of amorphous structures. This is confirmed by the absence of a halo on the PXRD, which is one of the characteristics of amorphous substances. According to the DSC data, GML-3 begins to melt at 87.13 °C with an endothermic peak of 89.71 °C and a release of 95.56 J/g (Figure 3f). The thermogram shows a melting peak characteristic of a crystalline substance.

Thus, the API molecules during synthesis assemble into crystals with a melting point of 89.71 °C, which agglomerate into particles with an average size of 58.64 µm. It is difficult to determine the type of the Bravais lattice and calculate the Miller indices for GML-3since the powder is studied, and growing a single crystal is a nontrivial task. However, the PXRD method in the case of GML-3 is well suited to control the level of crystallinity and the degree of change in the crystal lattice. The high degree of crystallinity and large particle size adversely affect the dissolution rate; therefore, despite satisfactory results in terms of particle sizes (virtually no particles larger than 200 µm), additional micronization of GML-3 is required to control the solubility/dissolution rate.

### 3.2. Micronization by the RESS Method

To select a micronization technology using supercritical fluids—RESS or SAS micronization of GML-3—a study of the solubility of API in CO_2_ was carried out (Table 2). Dissolution (or phase transitions in the GML-3-carbon dioxide system) was observed visually through the sapphire glass of the high-pressure cell as the system parameters changed: pressure and temperature.

After the complete dissolution of GML-3 at a pressure of 20 MPa and a temperature of 55 °C, CO_2_ was sprayed into the expansion chamber, and the resulting powder was collected. According to SEM before and after RESS, the average particle size had dropped from 58.64 µm to 30 nm, i.e., the particle diameter was reduced by approximately 2000 times (Figure 4a–f). The laser diffraction method for estimating the particle size distribution, in this case, is inappropriate to use since the GML-3 particle size is at the lower limit of the sensitivity of the method. For sizes less than 50 μm, it is necessary to use the Mie theory, which is impossible due to the lack of reliable data on the coefficients of transmission and refraction for GML-3. Notable is the color change in GML-3 from light yellow to light green.

PXRD data revealed several features indicative of a polymorphic conversion of API (Figure 5). Thus, for the particles obtained by sputtering from CO_2_, almost all PXRD peaks characteristic of the original GML-3 are absent, and new ones are present at 7.4°, 7.6°, 9.8°, 10.8°, 10.9°, 20.0°, and 22.5° (Figure 3 after RESS). According to the obtained results, the presence of a previously unexplored polymorphic form II of GML-3 can be seen. The single crystal structure could not be determined due to the difficulty of growing GML-3 single crystals suitable for PXRD studies. Throughout this section, the original API GML-3 is referred to as Form I, and the recrystallized RESS API GML-3 is referred to as Form II.

For additional comparative research on the study of polymorphic forms of GML-3, various well-established analytical (in addition to PXRD) methods were used, including FTIR spectroscopy, Raman spectroscopy, and DSC [38]. According to the DSC study, the melting point for Form I is 3 degrees higher than for Form II, i.e., Form II differs slightly from Form I in terms of thermal stability (Figure 6a,b). The more stable form of the API should always be used to avoid transformations during manufacture, delivery, or storage that could ultimately affect the properties of the drug and lead to undesirable therapeutic consequences [39]. The same difference in melting point was observed for polymorphs of various APIs described by Laszcz et al. and in the work of Machado Viana et al. [36,40].

When comparing different series of GML-3 substances (Form I) with Form II, it can be concluded that the change in the melting point is explained not by fluctuations in the size of particles and crystals in the samples but by the presence of another polymorphic form of API (Figure 5a). A similar pattern of endothermic processes was observed by Yao et al. [35]. When the heating rate at DSC was changed from 5 °C/min to 1 or 20 °C/min, the difference of 2–4 degrees was preserved (Figure 5b).

The DSC curves of Forms I and II showed a single endothermic effect indicative of the melting point. No other phase transitions were observed during DSC measurements during heating [36].

According to the heat of fusion rule, the thermal parameters indicate that Forms I and II are monotropically bound [36]. In addition, the difference in heat of fusion (heat of transition) is negative from the low-temperature form to the high-temperature form, indicating a monotropic transition between Forms I and II. Using IR spectroscopy, it was confirmed that the sample studied after RESS was not a product of GML-3 degradation (Figure 6a). When comparing the IR spectra, there are differences (the absence of a small peak for Form II) at wavelengths 1400–1450, 124, 940–890, and 575–500 cm^−1^.

For Raman spectroscopy after RESS, an additional high-intensity peak is observed at 650–550 cm^−^^1^ (Figure 6b). There are also differences at 700–800, 1250 cm^−^^1^, and more fluctuations at 1600–1150 cm^−^^1^. This method of analysis reliably distinguishes crystalline Forms I and II of GML-3. Thus, the presence of polymorphism in GML-3 obtained by the RESS method was proved. The obtained particles of GML-3 Form II have a particle size of 20–40 nm, which can positively affect the dissolution rate of the API. However, separate pharmacological studies are required for drug development.

### 3.3. Micronization of GML-3 by Dry Grinding

As a result of the dry grinding of GML-3, it was possible to micronize the particles to a size of 20–40 µm. Peaks characteristic of API with a decrease in their intensity and melting temperature are also preserved, which indicates the preservation of the crystalline form of particles (Form I). However, some researchers have expressed doubts about milling, as it can cause undesirable reactions in the API, affecting its physicochemical stability [41,42,43].

Thus, as a result of micronization, GML-3 particles acquired a residual static charge when working with them, which makes their use in any technological process difficult. Also, during storage, agglomeration of crystals was noticed, as a result of which the particle size increased over time. To test the stability of micronized GML-3, an accelerated aging test was performed. GML-3 before and after dry grinding was placed in an oven and thermostated for 175 h at a temperature of 55 °C. The original API retained its properties (particle size and size distribution), while the pre-micronized GML-3 agglomerated into large crystals.

According to the SEM and Fraunhofer laser diffraction data, the particles have an irregular shape, and the particle size distribution of the powder has the form of a Gaussian curve. More than 84% of the particles have a diameter greater than 250 µm, with individual particles up to 828.5 µm present (Figure 7). The powder itself is composed of agglomerates of geometrically irregular shapes consisting of crystals.

Thus, micronization by coarse mechanical grinding gives a negative result because the resulting agglomerates are crystals with a size several times greater than the diameter of the particles of the original API. The problems of agglomeration, residual static charge, and large particle size variation after micronization can be solved by adding a polymer, which will simultaneously increase solubility due to the solubilizing properties of the polymer. Composites of GML-3 and polymer co-dissolving in ethanol can also drastically increase the dissolution rate and inhibit crystallization. GML-3 is 1:7 soluble in ethanol, which evaporates quickly and has low toxicity. These alternative approaches to increasing solubility will be discussed next.

### 3.4. Dry Grinding with Polymer

The problem of agglomeration of GML-3 particles after mechanical grinding can be solved by adding, during micronization, a polymer used in the manufacture of drugs as a binder. This can prevent particle agglomeration and the buildup of static charge in the system. It may also lead to an increase in the dissolution rate due to the potential role of the polymer as a co-solvent in the water–polymer–API system.

Two polymers were selected: PVP and HPMC, which are highly soluble in water. When rubbed at a ratio of API–polymer 1:10 when working with samples, no phenomenon of static charge accumulation was observed. For a more complete description of the grinding with polymer method, the results of micronization of various APIs with PVP and HPMC derivatives were taken as a basis [44,45,46]. According to these studies, the polymer changed the size and morphology of the particles, making the API particles more rounded. However, for GML-3, the disappearance of the residual charge was observed only at a ratio of 1:10. This does not allow us to reliably estimate the particle size of GML-3 by laser diffraction and SEM. The GML-3 particle size data will overlap with the polymer data, and the laser diffraction method is poorly applicable for particles less than 50 microns. The small difference in the sizes of the initial API and polymers (the average particle size of PVP is 95 µm) and the low content of GML-3 in the mixture make it impossible to accurately estimate the size and morphology of GML-3 particles. For this reason, based on the experience of other studies on the joint micronization of API with polymer and the limitations of the dry grinding method, it is assumed that the particle size of GML-3 is comparable to the original API [14,15,16,17]. It is also assumed that the number of particles larger than 100 µm is an order of magnitude smaller than that of the original API. In this case, it is most important to evaluate the changes in the melting temperature and the degree of crystallinity of the mixture of GML-3 with polymer in comparison with the initial GML-3. Micronization with polymers can affect the dissolution kinetics not only due to the particle size but also by reducing the degree of agglomeration and crystallinity of API, the solubilizing properties of polymers, and the weak interaction between polymer particles and GML-3 [44,45]. In some cases, amorphization of API molecules is possible.

DSC data for a mixture of HPMC and API are characteristic of a two-component substance. First, GML-3 melting occurred, as evidenced by the characteristic DSC peak, then the HPMC glass transition process took place. These processes occurred as a whole, independently of each other. The PXRD data show GML-3 characteristic peaks superimposed on amorphous halos created by HPMC. Even though the polymer makes up more than 90% of the sample mass, the PXRD captures all the peaks characteristic of GML-3, which indicates the complete preservation of the API crystal lattice. Thus, a fully crystalline sample, in theory, can have a higher GML-3 rate/solubility due to the solubilizing properties of HPMC and the reduced agglomeration of API particles. When in water, HPMC will prevent GML-3 particles from sticking together into large conglomerates. According to the PXRD and DSC data, when crushed using PVP, GML-3 completely retained its crystallinity (Figure 8c,d). PVP, with a melting point above 150 °C, did not affect the parameters of the GML-3 crystal lattice or the shape of the DSC curve during grinding (Figure 8c,d). The PXRD data indicate the preservation of peak characteristics of GML-3 crystals. A decrease in the intensity of peaks indicates the grinding of crystals. However, even though the GML-3 crystal lattice as a whole has been preserved since PVP and HPMC have solubilizing properties, it is logical to conduct additional studies of the kinetics of dissolution of the obtained samples. The influence of the crystal lattice and agglomeration are further investigated. For this purpose, PVP mixtures with GML-3 were created by the solvent removal method. After drying API and polymer dissolved together in ethanol, we obtained samples with a reduced degree of crystallinity of GML-3 compared to mixtures. However, in this case, it is impossible to control the particle size inside the obtained samples.

### 3.5. Creating GML-3 and Kollidon 25 Composite

As an alternative to dry grinding with a polymer, the possibility of creating a bottom–up composite with the addition of PVP to the initial solution was studied. PVP was chosen as the polymer since, unlike HPMC, PVP is readily soluble in ethanol, like GML-3. The presence of the polymer during the solvent removal process may prevent the crystallization of GML-3, and the formation of a composite (eutectic mixture) will increase the solubility and stability of the API. GML-3 and PVP were dissolved in ethanol at an API:polymer ratio of 1:1,2,4, followed by the removal of the solvent.

The resulting samples were examined by PXRD and DSC methods for crystallinity. During the drying process, the polymer may begin to interfere with the crystallization of GML-3. In this case, a composite (eutectic mixture or ASD) is formed with a reduced peak intensity at PXRD (Figure 8e). The thermogram of the composite should exclude the melting peak characteristic of GML-3 (Figure 8f).

According to the data presented, at a ratio of 1:1.2, the samples obtained are not composites but a heterogeneous mixture of GML-3 and PVP with the preservation of GML-3 crystallinity. This is evidenced by the complete preservation of all characteristic GML-3 peaks in PXRD (Figure 8e). The thermogram also remained characteristic of GML-3 crystals (Figure 8f). Also, when working with samples, an accumulation of electrostatic charge was detected, which makes their use difficult to create a drug. This charge accumulation causes the mixture to adhere to the equipment and can result in the rapid recrystallization of GML-3, which will have negative consequences.

At a ratio of 1:4, eutectic was observed (according to DSC data) with a partial loss of crystallinity (Figure 8f). According to the DSC data, there is no peak melting of GML-3 crystals. This thermogram is more typical for a partially amorphous substance with signs of residual crystallinity. GML-3, in this case, is a kind of nanocrystal where the formation of large crystals is prevented by PVP. As a result, when heated, a modified thermogram is observed in the composite, which is more characteristic of the glass transition process. The melting/glass transition temperature decreased and became common for APIs and polymers. However, the residual crystallinity contributes to the thermogram since it cannot be described by a decreasing function. The increasing area on the thermogram indicates the residual crystallinity inside the composite. According to the PXRD data, the peak characteristics of GML-3 are almost completely absent (Figure 8e). The resulting composite is an eutectic mixture and may have a higher dissolution rate due to the disturbed crystallinity of GML-3. It can be assumed that with a further increase in the polymer content of the composite on the PXRD, the GML-3 peaks will disappear completely. This behavior is typical for ASD.

The solubility of the resulting GML-3/PVP composite was further studied; however, it is worth noting that residual crystallinity and relatively low electrostatic charge accumulation (compared to 1:1 and 1:2) may cause long-term storage problems in the form of agglomeration and recrystallization. This problem can be solved by creating composites of the second type—amorphous solid dispersions with various water-soluble polymers, followed by a study of their temporal stability; however, such experiments are beyond the scope of this study.

### 3.6. Dissolution Kinetics of GML-3 in Water

All previously obtained samples were used as objects for research. This study was conducted in various environments (water, water–alcohol solutions, 0.01 M HCl, acetate buffer) (Figure 9). Data on the kinetics of dissolution of GML-3 in distilled water allowed us to evaluate the effect of the use of various grinding methods on the solubility of GML-3. However, the aggregation of particles in contact with water and the crystallinity of GML-3 have a strong effect on the result. GML-3 (Form I) and GML-3 (Form II), as well as composites based on them, were additionally investigated in a water/ethanol mixture to assess the effect of each of the factors on solubility. The presence of ethanol improves the solubility of GML-3 and reduces the degree of aggregation of particles after entering the solution. The issues of using a mixture of solvents in the preparation of drugs from GML-3, as well as the solubility of GML-3 when samples enter the gastrointestinal tract, were discussed. The obtained data are shown in Figure 9.

According to the data presented, Form II has a higher solubility than Form I (by a factor of 430), but it is also characterized as practically insoluble in water (Figure 9a). The solubility of GML-3 after dry grinding remained unchanged compared to the original API. This may be due to the accumulation of residual electric charge, as a result of which the agglomeration of particles occurs both after grinding and after entering the aqueous medium. Grinding with polymer significantly increased the release of GML-3 in distilled water, increasing the solubility of GML-3 when milled with PVP by 800 and 400 times for HPMC. Bottom–up mixtures of GML-3 and PVP at ratios of 1:1 and 1:2 increased the solubility of GML-3 (60 min after the start of the test) by 450 and 800 times, respectively. The resulting composite GML-3-PVP (1:4) provided the highest solubility/dissolution rate of GML-3 in water—60 min after the start of the test, 80% of the API was dissolved in water, which is 1600 times higher than the original API (Form I).

In the technology of obtaining dosage forms, co-solvents are often used, which makes it possible to achieve API dissolution and use the solution with wet granulation. The tablet is the most common dosage form (DF). It is planned to use GML-3 in the future to create GML-3 tablets with anxiolytic and antidepressant effects. The technology of composite production, in this case, can be scaled to production using a mixture of solvents (ethanol–water). PVP is highly soluble in water and ethanol. GML-3 and PVP can be dissolved in alcohol and then added to water until a 50% solution is obtained (GML-3 precipitates at an alcohol strength of less than 34%). This will allow you to leave GML-3 in dissolved form as well as preserve the binding properties of PVP.

The effect of the ethanol percentage on the dissolution kinetics of GML-3 is shown in Figure 9b–d. The experiments were carried out in 30, 50, and 70% alcohol. In 30% ethanol, the samples dissolved faster due to the solubility of GML-3 in ethanol and less agglomeration of particles (Figure 9b). GML-3 was released from the composite (1:4) at a higher rate, but the total solubility remained at 80%. It is worth noting that GML-3 (Form I) dissolved by 30%; moreover, small particles dissolved and large ones remained in the form of sediment. The dependence of the solubility of GML-3 on the particle size is difficult to estimate since large particles can be both crystals and agglomerates of crystals. Moreover, when ingested into aqueous media, agglomeration of particles is observed, which greatly affects the result. For the ratio GML-3:PVP = 1:2, the solubility was 50%, which was 1.5 times higher than the solubility of GML-3 (Form I) in a similar solvent mixture. This could be facilitated by the solubilizing properties of the polymer and the reduced aggregation of particles when the sample enters the solution due to the presence of PVP between the particles. The strong influence of aggregation and crystallinity on the dissolution kinetics is indirectly confirmed by data on the dissolution kinetics of GML-3 (Form II) and samples with a ratio of PVP:GML-3 = 1:1. Thus, although GML-3 (Form II) has a particle size of 20–40 nm, only half of GML-3 dissolves in 30% ethanol. Recrystallized with PVP GML-3 (1:1) is characterized by the preservation of the crystal lattice, and PVP in such a concentration did not prevent crystallization and did not prevent agglomeration of particles when entering the solution.

In 50% alcohol, the degree of agglomeration of particles in contact with water is reduced. GML-3 (Form II) has a high dissolution rate/solubility (94%). This is due to an increase in alcohol content and a small particle size (Figure 9c). GML-3 (Form I) is characterized by solubility at the level of 70%. Approximately the same solubility was possessed by GML-3 mixed with PVP. This may indicate that, in this case, the crystal structure of GML-3 prevents greater dissolution. For the composite (1:4), an increase in the solubility of GML-3 (88%) was observed, which may be associated with the solubilizing property of PVP and with a decrease in the degree of crystallinity of GML-3. However, complete dissolution is not observed due to the fact that GML-3 has partially retained its crystal lattice. For samples where PVP:GML-3 = 1:2, the dissolution rate/solubility is lower even than that of GML-3 (Form I) due to the fact that PVP forms larger particles together with GML-3. For this reason, when interacting with the solution, some of the particles agglomerate and precipitate, slowing down the dissolution.

For 70% ethanol, the kinetics of dissolution of polymorphic forms of GML-3 remained intact (Figure 9d). This suggests that it is the crystallinity of GML-3 that prevents the complete dissolution of GML-3. Thus, a partial change in the degree of crystallinity in the composite allowed for an increase in the degree of release to 87%.

It is worth mentioning that in alcoholic solutions, large crystals and agglomerates of GML-3 particles usually remain in the undissolved sediment. These data suggest that it is impossible to achieve complete dissolution of GML-3 in water only by grinding the API. Creating a polymorphic form of GML-3 also does not completely solve the problem. The problems associated with the aggregation of particles in the medium and the energy of the crystal bond can be further solved by creating composites where GML-3 will be completely in an amorphous state. This type of composite is often called amorphous solid dispersion. This is achieved by increasing the polymer content in the composite. As a result, in the composite, GML-3 represents disconnected molecules whose crystallization is prevented by the polymer. However, there are some difficulties. The resulting composite should, according to DSC and PXRD data, show no signs of residual crystallinity and have the stability of an amorphous state for a long time. An additional advantage will be the increased content of the polymer, which can increase the solubility of the API. However, the creation of these polymers is beyond the scope of our research.

We have considered the solubility of samples in hydrochloric acid (stomach medium) since it is assumed that DF will be for oral administration. Thus, GML-3 (Form I) and GML-3 (Form II) showed a solubility of more than 90%, which GML-3 (Form I) achieved in 60 min and GML-3 (Form II) in 10 min (Figure 9e). We can talk about the pH-dependent solubility of GML-3, which makes the oral route of administration more promising. All samples showed good solubility in hydrochloric acid, and the composite (1:4) demonstrated the highest dissolution rate (5 min). However, possible pH fluctuations can have a strong effect on the solubility of the API. In a state of fasting, the stomach usually contains about 10–50 mL of acidic contents (pH 1–2) [47,48,49]. The introduction of API together with 240 mL of water leads to a pH change of 4.6 [50,51]. Eating also leads to a change in the pH. The acidity is subsequently restored, but the rate of its change can be different and take from 30 min to several hours. At pH = 4.6, GML-3 (Form I) and GML-3 (Form II) have reduced solubility compared to hydrochloric acid (30% and 50%). Only the composite (1:4) retains a high level of GML-3 release. Thus, the eutectic mixture (1:4) demonstrated the best solubility in all media. Thus, the complete rapid dissolution of GML-3 can be ensured in the future by the creation of amorphous solid dispersions and tablets based on them.

## 4. Conclusions

The results of this study demonstrate the effect of various grinding methods and the creation of composites with PVP and HPMC on the solubility/dissolution rate of GML-3 in water. Practically insoluble in water, GML-3 is a powder with an average particle size of 58.64 µm, consisting of agglomerates of crystals with a melting point of 87.13 °C. GML-3 dissolves in CO_2_ at a temperature of 55 °C and a pressure of 20 MPa, which made it possible to spray the GML-3 solution through a 1.5 mm nozzle into an expansion chamber to obtain particles with a size of 20–40 nm, which is 2000 times smaller than the original API. It should be noted that the particles obtained by this method were a previously unstudied polymorphic form of GML-3, the presence of which was proved in this study. A different crystal lattice from the original API and a smaller particle size give the GML-3 polymorph 430 times greater solubility (after measurement after 60 min). Mechanical grinding did not affect the dissolution rate of GML-3. Moreover, after storage of the ground API, particle agglomeration was observed, which led to a sharp increase in particle size to more than 250 μm. Co-grinding with PVP and HPMC increased the solubility by 800 and 400 times. Bottom–up mixtures of GML-3 and PVP at ratios of 1:1 and 1:2 increased the solubility of GML-3 by 450 and 800 times, respectively. The composite was obtained at a minimum ratio of GML-3 to PVP of 1:4, with a 1600-fold increase in solubility, which is the best result among all the studied samples.

## Figures and Tables

**Figure 1 polymers-15-04136-f001:**
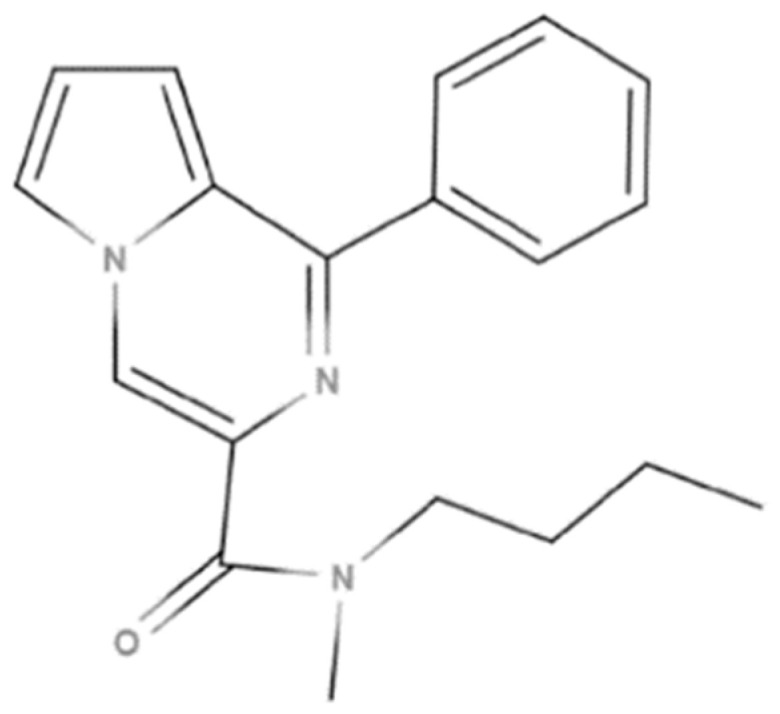
Structural formula of GML-3.

**Figure 2 polymers-15-04136-f002:**
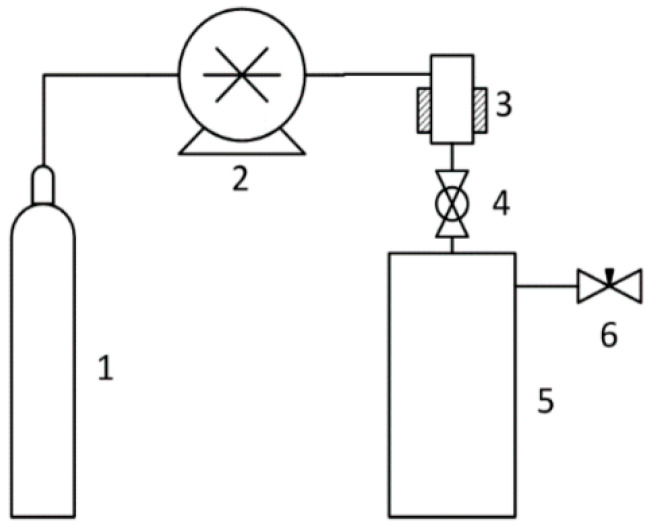
Scheme of a device for rapid expansion of supercritical suspensions: 1—CO_2_ source; 2—CO_2_ pump; 3—suspension chamber; 4—spray nozzle with a diameter of 1.5 mm; 5—atmospheric pressure sedimentation tank; and 6—valve.

**Figure 3 polymers-15-04136-f003:**
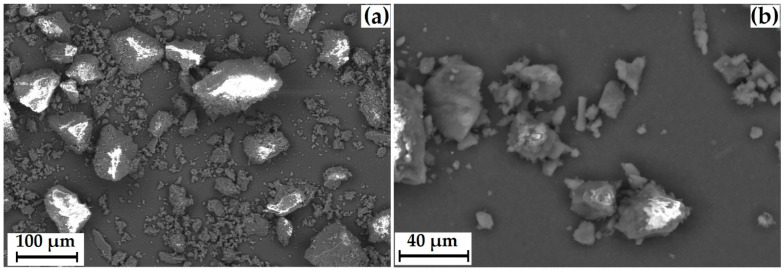
GML-3 data: SEM of particles at magnification ×264 (**a**), ×1200 (**b**), ×2300 (**c**), laser diffraction (**d**), PXRD GML-3 data before and after RESS (**e**), and DSC (**f**). Black line—GML-3 (Form I), green line—GML-3 (Form II), red line (laser diffraction)—differential curve, red line (DSC)—temperature change, blue line (laser diffraction)—integral curve, light blue (DSC)—peak area. The thermogram (DSC) of the GML-3 melting process shows the area of the process, the temperature of the start, peak and end of the process, and the amount of energy released.

**Figure 4 polymers-15-04136-f004:**
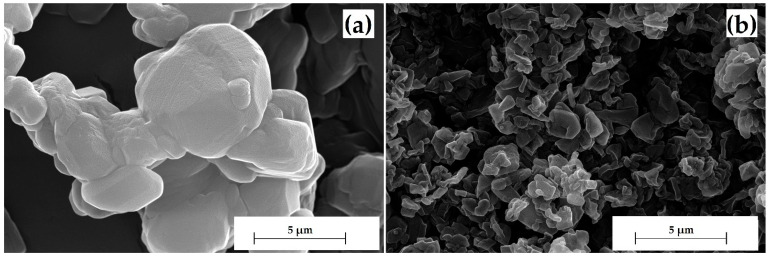
SEM of GML-3 particles prior to dissolution in CO_2_ at resolutions ×24,000 (**a**) ×10,000 (**c**) ×500,000 (**e**) and for particles obtained after RESS ×24,000 (**b**) ×10,000 (**d**) ×500,000 (**f**).

**Figure 5 polymers-15-04136-f005:**
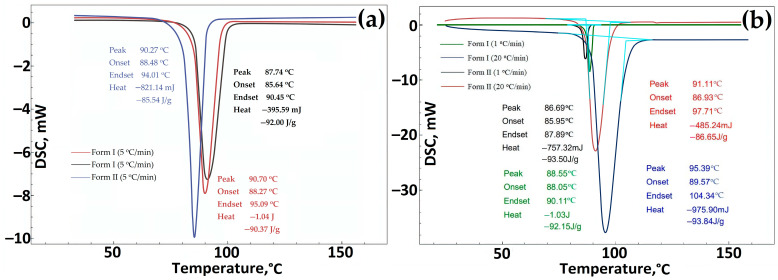
Data on DSC GML-3 before RESS (Form I) and after recrystallization by the RESS method (Form II). The heating rate was 5 °C/min (**a**), 1 °C/min (**b**), 20 °C/min (**b**). Light blue—peak area.

**Figure 6 polymers-15-04136-f006:**
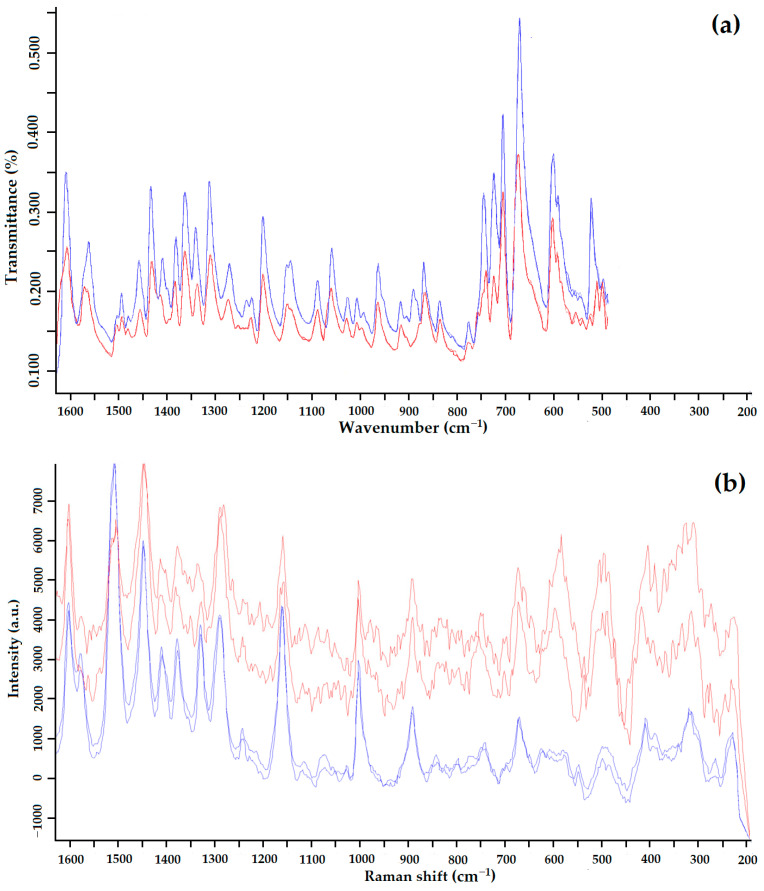
IR spectrum (**a**) and Raman spectrum (**b**) of GML-3 Form I (blue) and GML-3 Form II (red).

**Figure 7 polymers-15-04136-f007:**
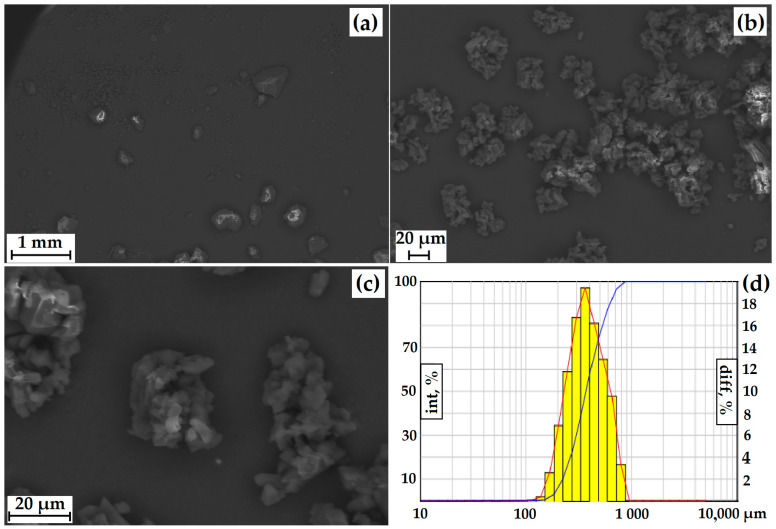
SEM of GML-3 particles after micronization and accelerated aging storage at ×50 (**a**) and ×778 (**b**), ×2400 (**c**), and laser powder diffraction data (**d**). The red line—differential curve, the blue line—an integral curve.

**Figure 8 polymers-15-04136-f008:**
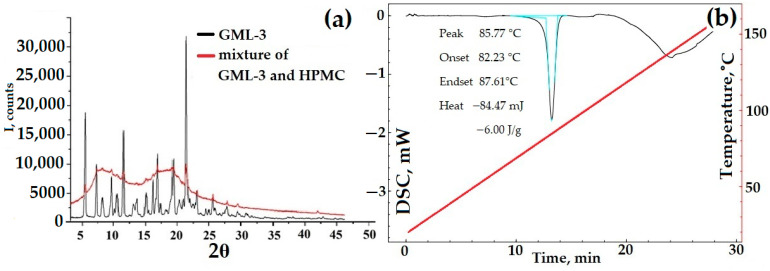
PXRD and DSC data for mechanical blending of GML-3 with HPMC (**a**,**b**), GML-3 with PVP (**c**,**d**) compared to GML-3, and composites of GML-3 with PVP (**e**,**f**). Light blue (**b**,**d**)—peak area, red line—temperature change. The thermogram (**b**,**d**) of the GML-3 melting process shows the area of the process, the temperature of the start, peak and end of the process, and the amount of energy released.

**Figure 9 polymers-15-04136-f009:**
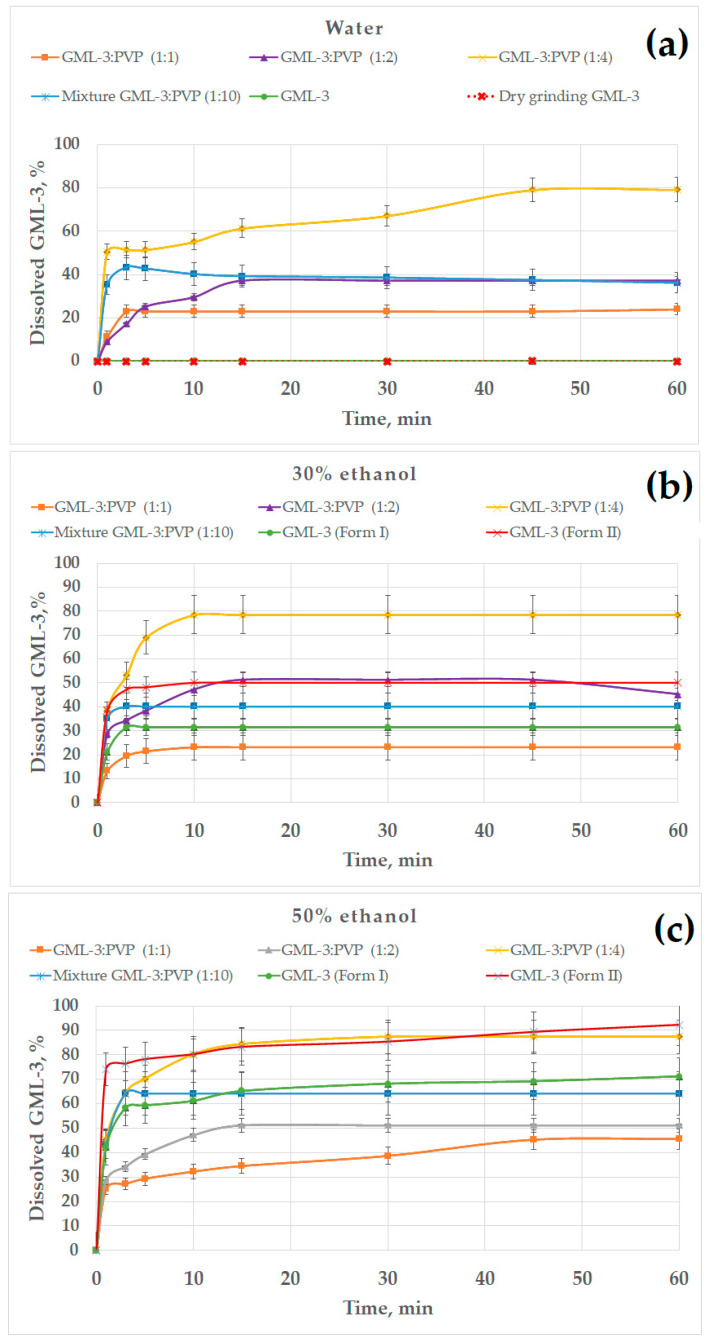
The dissolution kinetics of different GML-3 formations in distilled water (**a**); 30% (**b**), 50% (**c**), and 70% ethanol (**d**); 0.01 M HCl, simulating gastric juices in the stomach (**e**); and an acetate buffer, simulating the upper intestine conditions (**f**).

**Table 1 polymers-15-04136-t001:** Physicochemical properties of the GML-3 compound.

Properties	Value
Molecular mass (M, Da)	307.39
Melting temperature (Tm, °C)	87–89
Solubility in water	More than 1:10,000
Solubility in ethanol	1:7

**Table 2 polymers-15-04136-t002:** Data on the solubility of GML-3 in CO_2_ at different pressures and temperatures.

Parameters	Visual Observation Data
T= 45 °C; P = 5 MPa	no dissolution
T= 45 °C; P = 10 MPa	no dissolution
T= 45 °C; P = 15 MPa	no dissolution
T= 45 °C; P = 20 MPa	no dissolution
T= 55 °C; P = 5 MPa	no dissolution
T= 55 °C; P = 10 MPa	no dissolution
T= 55 °C; P = 15 MPa	partial dissolution
T= 55 °C; P = 20 MPa	full dissolution

## Data Availability

Not applicable.

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
