# Peer review of "Modeling of the Aqueous Solubility of N-butyl-N-methyl-1-phenylpyrrolo[1,2-a] pyrazine-3-carboxamide: From Micronization to Creation of Amorphous–Crystalline Composites with a Polymer"

_polymers, 2023, doi:10.3390/polym15204136_

Round 1
Reviewer 1 Report (Previous Reviewer 1)
The authors responded to the comments. The revised version appears, in my opinion, ready to be accepted for publication.
Author Response
2023-10-11
Resp. to Rev.1
Dear Reviewer:
Thank you so much for your evaluation of our humble contribution.
Regards
Dr. Alex Vetcher
Reviewer 2 Report (Previous Reviewer 2)
Thank you again for opportunity to review the revised manuscript entitled "Modeling of aqueous solubility of N-Butyl-N-methyl-1-phenylpyrrolo[1,2-a] pyrazine-3-carboxamide: from micronization to creation of amorphous-crystalline composites with a polymer." Unfortunately, I failed to understand the clear novelity and scientific soundness in the story the authors prepare. For acceptance in this or other journal, introduction section, presentation of data, and result section may need to be elavorated. Especially, introduction section because it was confusing to understand the purpose and novelity. This article can be misinterpreted as a scientific report rather than scientific journal without major modifications.
Author Response
2023-10-11
Resp. to Rev.2
Dear Reviewer:
Thank you so much for your time and efforts to improve our contribution.
In your comments you underlined that you “failed to understand the clear novelity and scientific soundness in the story the authors prepare.” In our latest version we tried to improve both points.
Please let us know what do you think about it.
Regards
Dr. Alex Vetcher
Reviewer 3 Report (Previous Reviewer 3)
The manuscript cab be accepted in present form.
Author Response
2023-10-11
Resp. to Rev.3
Dear Reviewer:
Thank you so much for your evaluation of our humble contribution.
Regards
Dr. Alex Vetcher
Reviewer 4 Report (New Reviewer)
How did the author ensure the results of particle size? especially, the obtained particles are irregular in shape?
The obtained Gaussian distribution of particles might be poorly correlated with the obtained particle size displayed in SEM graphs.
In the dissolution test, the authors mentioned that they measured the release in water. However, in the results, they mentioned acetate buffer and ethanol
How did the authors choose the wavelength which they measure the dissolution test?
Moderate English language is recommended
Author Response
2023-10-11
Resp. to Rev.4
Dear Reviewer:
Thank you so much for your time and efforts to improve our contribution.
We addressed all the comments in the order in your review:
- How did the author ensure the results of particle size? especially, the obtained particles are irregular in shape?
We explain it in the body of the submission
- The obtained Gaussian distribution of particles might be poorly correlated with the obtained particle size displayed in SEM graphs.
It was a technical mistake. In novel version it looks improved. Laser diffraction let to measure a large population, while SEM only a few dozens. We also changed “conditionally correct” to asymmetric.
- In the dissolution test, the authors mentioned that they measured the release in water. However, in the results, they mentioned acetate buffer and ethanol
Corrected
- How did the authors choose the wavelength which they measure the dissolution test?
We added necessary corrections in the body.
Please let us know what do you think about it.
Regards
Dr. Alex Vetcher
Round 2
Reviewer 2 Report (Previous Reviewer 2)
Thak you again for the opportunity to review the article.
With some additional description in the manuscript, the needs of research were understandable. The revised version was much better than original version. However, there are still some points that must be improved as follows:
1. Too long introduction section should be summarized only with necessary information to understand the purpose and approach in this research.
2. In Line 479, the authors described that "The particle size obtained from the combined grinding of polymer and API is identical to that obtained from dry grinding of pure API." Please attach figures in the manuscipt.
3. In Line 537, the author described that ". This thermogram is more typical for a partially amorphous substance with signs of residual crystallinity." Why were only composites of GML-3 with PVP (1:4) eutentic mixture?
Author Response
2023-10-13
To Rev.2
Dear Reviewer
Thank you again for your efforts to increase readability of our submission.
Let us list our responses in their order in your review:
- Too long introduction section should be summarized only with necessary information to understand the purpose and approach in this research.
Corrected.
- In Line 479, the authors described that "The particle size obtained from the combined grinding of polymer and API is identical to that obtained from dry grinding of pure API." Please attach figures in the manuscript.
The idea was clarified.
3. In Line 537, the author described that ". This thermogram is more typical for a partially amorphous substance with signs of residual crystallinity." Why were only composites of GML-3 with PVP (1:4) eutectic mixture?
The description of thermogram was elaborated with the clear idea of our consideration.
Please let us know, what else can be done to make our submission more attractive to the readers.
Sincerely
Dr. Alex Vetcher
Reviewer 4 Report (New Reviewer)
The authors have covered all the raised questions
Minor English editing is required
Round 3
Reviewer 2 Report (Previous Reviewer 2)
Thank you again for the opportunity to review this article. The manuscript is qualified enough to be accepted in Polymers, I think. Story is understandable by readers, and the data included in the manuscript support the story.
This manuscript is a resubmission of an earlier submission. The following is a list of the peer review reports and author responses from that submission.
Round 1
Reviewer 1 Report
This is an interesting report on different physicochemical approaches to increasing the water solubility of GML-3 (N-Butyl-N-methyl-1-phenylpyrrolo[1,2-a]pyrazine-3-carboxamide]). GML-3 is an 18 kDa mitochondrial translocator protein (TSPO) ligand and a potential candidate for combination drug therapy due to its anxiolytic and antidepressant activity. Lack of most of the side effects comparing to diazepam makes it a good candidate for drug. However, its low water solubility and tendency to agglomerate prevent its release. The goal of presented study was the investigation of dry grinding effects, rapid expansion of a supercritical solution (RESS) and eutectic mixture (composite) of GML-3 with polyvinylpyrrolidone (PVP) on the particle size, dissolution rate and lattice retention of GML-3.
· My general concept comments:
1) Since the problem of insolubility has not been solved, in which pharmaceutical form can GML-3 be used?
2) As is known, in the technology of drug forms, the dissolution of the active substance in the presence of co-solubilizers is used. Perhaps it would be good to study the dissolution kinetics of the selected GML-3/polymer systems in the water/ethanol medium system.
3) The discussion of the results is cursory and not very deep. The article would be much appreciated if it was corrected.
· My detailed remarks are the following:
1. Are any pseudopolymorphs of GML-3 known?
2. Page 2, line 64: There is a mistake in size of API particles. According to cited ref 7: “The traditional size of pharmaceutical powders, generally ranging from a few hundred microns up to 50 μm”
3. Page 4, line 150: please provide a specification of Mn and PD - this is very important for characteristics of polymers.
4. Page 6, line 234: PXRD and DSC data provide different information so it should be discussed separately.
5. Fig 6a: DSC curves of form I – two different curves are at the same heating rates, is it correct?
6. Generally, results are presented a bit carelessly, scale font, axis descriptions should be unified. In some places even the description of the axis or units is missing (e.g. Fig. 9b, the right y axis) or number format should be corrected (Fig 7a – y axis).
7. Page 11: chapter 3.4 Dry grinding with polymer - the discussion on the influence of the polymer on the obtained effect should be extended. How repeatable is the process?
8. Fig 9 a, c – mechanical mixture should be named simply a “mixture”.
9. Page 13, line 407 and 411: The description of the API to polymer ratio is unclear.
10. Fig 10a: standard deviation is invisible for orange curve – is it so small that it is obscured by the points of the curves?
.
Reviewer 2 Report
Thank you for great opportunity to review the article entitled "Modeling of aqueous solubility of N-Butyl-N-methyl-1-phenylpyrrolo[1,2-a]pyrazine-3-carboxamide: from micronization to creation of amorphous-crystalline composites with a polymer." The authors evaluated particle sizes, crystallinity, IR spectrums for samples prepared by different methods; however, data presentation and results/discussion must be improved for acceptance.
Comments:
1. Introduction section must be structured in order to provide readers with the importance and/or novelity of importance. I recommend you to modify the introduction section to meet the point.
2. The related figures should be combined in one figure. (For example, Fig.3f and Fig.5 show the same data on GML-3.)
3. The authors should make explanation not only on data but also on why the results obtained and/or what factors have an major impact. For example, the authors should give more discussion in dissolution kinetics of GML-3, the author's research focus.
After modification, re-submission can be considered.
I hope this paper would be accepted in other journals.
Reviewer 3 Report
In this study, the authors used several methods to change the particle size and the crystal texture, resulting in improve the solubility of GML-3. These methods will improve the solubility and bio-availability of GML-3. These methods show good economic and social benefits. However, all the used methods are common process, and the polymer used in this manuscript (PVP) is common on the market. Therefore, the novelty is low.
1. The novelty of this manuscript need to highlight;
2. Some professional journals on pharmacy are more appropriate for this manuscript;
3. The relationship between particle size and the solubility of GML-3 need to be studied;
4. The language needs minor edition;
5. The release profiles of GML-3 from the GML-3/PVP composite need provide;
6. The influence of crystal texture of GML-3 crystals on their solution need to be discussed in detail;
7. The direction of heat flow should be added into the DSC curves;
8. The axis of abscissas in Fig. 7 a and b are not uniform;
Minor editing of English language required.